# Evaluating the Role of Data Enrichment Approaches towards Rare Event Analysis in Manufacturing

**DOI:** 10.3390/s24155009

**Published:** 2024-08-02

**Authors:** Chathurangi Shyalika, Ruwan Wickramarachchi, Fadi El Kalach, Ramy Harik, Amit Sheth

**Affiliations:** 1Artificial Intelligence Institute, College of Engineering and Computing, University of South Carolina, Columbia, SC 29208, USA; ruwan@email.sc.edu; 2McNair Center for Aerospace Innovation and Research, Department of Mechanical Engineering, College of Engineering and Computing, University of South Carolina, Columbia, SC 29201, USA; elkalach@email.sc.edu (F.E.K.); harik@mailbox.sc.edu (R.H.)

**Keywords:** rare events, event detection, event prediction, time series, data enrichment, smart manufacturing

## Abstract

Rare events are occurrences that take place with a significantly lower frequency than more common, regular events. These events can be categorized into distinct categories, from frequently rare to extremely rare, based on factors like the distribution of data and significant differences in rarity levels. In manufacturing domains, predicting such events is particularly important, as they lead to unplanned downtime, a shortening of equipment lifespans, and high energy consumption. Usually, the rarity of events is inversely correlated with the maturity of a manufacturing industry. Typically, the rarity of events affects the multivariate data generated within a manufacturing process to be highly imbalanced, which leads to bias in predictive models. This paper evaluates the role of data enrichment techniques combined with supervised machine learning techniques for rare event detection and prediction. We use time series data augmentation and sampling to address the data scarcity, maintaining its patterns, and imputation techniques to handle null values. Evaluating 15 learning models, we find that data enrichment improves the F1 measure by up to 48% in rare event detection and prediction. Our empirical and ablation experiments provide novel insights, and we also investigate model interpretability.

## 1. Introduction

Events are occurrences linked to particular locations (spatial), time frames (temporal), and contexts (semantics). Rare events, a subset of these, are notable for their infrequency. The degree of the infrequency of rare events is typically influenced by the specific field of application [1,2]. Rare events can be categorized into distinct categories based on factors like the distribution of data, significant differences in rarity levels, and the context: extremely rare events have a frequency of 0–1%, very rare events have a frequency of 1–5%, moderately rare events have a frequency of 5–10%, and frequently rare events have a frequency greater than 10% [3]. In real life, rare events can be observed ubiquitously in various industries such as earth science, manufacturing, telecommunication, healthcare, transportation, economies, and energy. They span various applications, including rare disease diagnosis, anomaly detection, fraud detection, and natural disaster prediction [3]. Rare events, notable for their scarcity, provide vital information in any domain and pose challenges including but not limited to the “Curse of Rarity” (CoR) [4]. This leads to a hindering of decision-making, modeling, verification, and validation due to the events’ exceptional rarity [1,2,4,5,6,7,8,9]. Rare events, by their intrinsic nature, exhibit significant challenges in the manufacturing domain. Some examples of rare events in manufacturing are paper breaks in the paper manufacturing industry, component failures in the automotive industry, supply chain disruptions in food processing and packaging, etc. Rare events can have a significant impact when they occur. These disruptions are costly for industries; resumption time is higher. For example, in pulp-and-paper manufacturing, paper breakage that occurs <1% of the time can cost the industry a huge loss [10]. Predicting such hard-to-predict occurrences is important for cost management, operational efficiency, and energy conservation. Identifying them leads to a reduction in defects, a lowering of equipment downtime, and the optimizing of energy consumption. Thus, the identification and anticipation of such events hold paramount importance for ensuring the overall productivity of the industry. The sporadic occurrence and often unpredictable nature of these events make their detection and prediction imperative. While detecting such events is significant, the ability to foresee them beforehand assumes greater importance, facilitating proactive measures to mitigate potential consequences. It would lead to ensuring optimization, quality, and safety standards in manufacturing processes.

There is a difference between the detection and prediction of rare events in any industry based on their occurrence. Detection involves identifying the occurrences of rare events after they arise, whereas prediction involves forecasting the likelihood of future occurrences of such events before they manifest. Detection emphasizes recognizing and flagging rare occurrences within use cases that deviate significantly from the norm or occur infrequently. Prediction revolves around anticipating, forecasting, or modeling the occurrence of infrequent events based on historical data and patterns.

In machine learning, the quality and richness of data play an integral role in the development and efficacy of predictive models. Data enrichment techniques emerge as pivotal methodologies employed to augment and refine raw datasets, infusing them with additional information, context, and structure. The core concept of data enrichment is to address the deficiencies present in raw datasets, including missing data points (incompleteness), scarce representations (sparsity), and underlying distortions or prejudices (inherent biases). Data enrichment benefits machine learning models, imparting them with enhanced capabilities to comprehend complex patterns, generalize effectively, and make more accurate predictions.

Common domain-agnostic data enrichment techniques include data augmentation, sampling, imputation, domain knowledge integration, data transformation, etc. Our study particularly focuses on data augmentation, sampling, and imputation. Data augmentation focuses on generating synthetic data to increase the input space [11,12,13]. In our study, we focused on increasing only the feature space of the data while keeping the number of data points across the time steps unchanged. Sampling techniques aim to balance the data distribution by synthesizing or eliminating the number of data points, which is a common technique used in imbalanced data analysis [14]. Imputation techniques focus on creating accurate and complete data by making adjustments for unspecified values in the data [15,16]. The amalgamation of these techniques offers an opportunity to leverage multivariate information and compensate for the scarcity of event instances. These would overall lead to enhancing the quality, diversity, and utility of the dataset for machine learning tasks.

To the best of our knowledge, none of the previous studies explored the potential of applying different data enrichment techniques using different classifiers in rare event prediction on real-world manufacturing datasets. This research investigates the application of statistical, machine learning, and deep learning-based modeling techniques with three data resampling techniques, eleven data augmentation techniques, and two data imputation techniques in predicting rare events.

The main contributions of this paper are listed below:

### 1.1. Primary Contributions

1.The primary contribution of this work is to propose a framework to investigate the role of data enrichment approaches in rare event detection and prediction (data augmentation, sampling, and imputation).2.We introduce a real-world dataset from a product assembly manufacturing industry.

### 1.2. Secondary Contributions

1.We conduct empirical and ablation experiments on five real-world datasets from the manufacturing domain for rare event detection and prediction to derive dataset-specific novel insights.2.We investigate the interpretability aspect of models for rare event prediction considering multiple methods.

The rest of the paper is organized as follows: the Related Works section presents previous studies on leveraging data enrichment techniques in rare event detection and prediction. The Methodology section provides a brief overview of the datasets adopted, the methods used for rare event detection and prediction, and the data enrichment approaches used in this study. The modeling results with their experiments have been included in the Evaluations section. It discusses the results of the proposed models and the model and feature analysis based on explainable learning methods. Finally, the Conclusions and future research section provides an overall summary of this research along with significant findings, limitations, and future scopes of research.

## 2. Related Works

In early studies, data enrichment techniques have been used to enhance existing datasets with additional information to improve the quality, diversity, and predictive power. Considering the scope of our research, we explore the related studies across three data augmentation techniques: data augmentation, sampling, and imputation. Data augmentation techniques expand dataset size and diversity, sampling methods address class distribution balance, and imputation strategies effectively manage null values within datasets. Data augmentation is a widely employed method in machine learning that involves expanding the size and diversity of datasets by creating additional data samples through various transformations while retaining the original labels. Time series data augmentation has received considerable attention in the literature [11,17]. With respect to rare event prediction, these techniques exhibit utility across a wide array of domains, including machinery fault diagnosis, computer vision, and geology. For instance, in a manufacturing-related use case [18], researchers introduced data augmentation by generating new features through Fast Fourier Transform (FFT) alongside the given features, substantially enhancing a multivariate time series dataset [10] in pulp-and-paper manufacturing. Other studies in rare event prediction have leveraged data augmentation techniques, including variations in Generative Adversarial Networks (GANs), conditional GANs (CGANs), and Wasserstein GANs (WGANs) [19]. These approaches have generated labeled samples for various applications, such as mineral prospectivity prediction and scene change event detection, using image-based data augmentation techniques like cropping geological image data and window-based augmentation [20,21,22]. These methods contribute to enhancing the robustness and effectiveness of rare event prediction models by enriching the available datasets.

Sampling techniques are integral to machine learning, especially when handling imbalanced datasets, such as those focused on anomalies, failures, and rare events. Rare event research has employed various categorizations for sampling methods, including both basic and advanced methods. Basic sampling techniques involve randomly undersampling the majority class or oversampling the minority class. In rare event research, these methods have been applied to balance class distributions [10,23,24,25,26,27]. Researchers have explored combining sampling techniques with clustering models [26,28], ensemble learning methods [29], advanced architectures like the Siamese CNN [21], and statistical sampling methods like Hoeffding bounds [30] to enhance predictive performance in rare event prediction. Yet, random sampling has drawbacks, including overfitting in oversampling and data loss in undersampling. Advanced sampling techniques, on the other hand, delve into more sophisticated strategies to intelligently balance classes. Widely applied methods include the Synthetic Minority Oversampling Technique (SMOTE) [19,23,24,31,32,33], which generates synthetic minority cases to address overfitting, and Adaptive Synthetic Sampling (ADASYN) [33,34], which provides weighted oversampling to tackle difficulty in learning. For example, it has been observed that logistic regression, when coupled with the SMOTE, has yielded better outcomes in the identification of Look-Alike, Sound-Alike (LASA) cases in textual data [24]. Furthermore, Asraf et al. [33] have leveraged ADASYN with the XGBoost model to facilitate rare event modeling in identifying high-risk segments in Wrong-Way Driving.

Other advanced sampling techniques used in rare event research include Edited Nearest Neighbor (ENN) [33,35,36], the Neighborhood Cleaning Rule (NCL) [36,37], NearMiss (NM) [33,36,38], and One-Sided Selection (OSS) [23,29,39,40,41], which focuses on reducing noise and refining decision boundaries. One-sided Selection (OSS) is an undersampling technique that intelligently removes noisy majority-class instances, using Tomek Links (TLs) and the Condensed Nearest Neighbor (CNN) rule [39], to create a more refined and representative subset of the majority class in rare event studies [23,29,40,41]. TL can remove noise and boundary points in majority class samples in rare events [33,36,41,42]. Furthermore, advanced methods such as Cluster-based Oversampling (CBO) handle both between-class and within-class imbalances simultaneously [23,25]. Time Series subsampling reduces the computational cost of analyzing long time series data by selecting a subset of time points from the original to create a new, shorter, and more balanced time series [43,44]. Stratified sampling ensures a proportional representation of classes in the sampled dataset by dividing the dataset into subgroups (or strata) based on the target variable’s classes and then randomly sampling from each stratum. In industrial machinery fault diagnosis [45], subsampling of time series data was employed to ensure a balanced representation of normal and different bearing fault types; also, it uses stratified techniques to appropriately represent these subgroups in the final dataset. Additionally, audio data sampling [46], uncertainty sampling [47], and choice-based or endogenous sampling [48,49] cater to specific data modalities and sampling goals, making them valuable in rare event prediction use cases.

Imputation techniques play a pivotal role in addressing missing or incorrect data values in datasets, ensuring their completeness and accuracy. These techniques can be categorized into two main types: simple imputation and advanced imputation. Simple imputation methods encompass straightforward strategies such as mean or median substitution, which replace missing values with the mean or median of the respective attribute [19,50,51,52]. Interpolation is another simple imputation approach that estimates missing values based on neighboring observed data points, considering underlying data trends [27,53]. These simple techniques have been used in predicting cancer survival among adults in the SEER dataset [50], predicting failures in the Air Pressure System (APS) dataset [19,51,52], and solar flare forecasting [27] for the imputation of missing values in rare-event time series prediction.

In contrast, advanced imputation methods delve into more complex approaches such as iterative imputation, multiple imputation, soft impute, expectation maximization, offset value approximation, and Singular Value Decomposition (SVD) imputation. These techniques provide a more sophisticated means of estimating missing values, utilizing statistical modeling and iterative processes. Advanced imputation approaches are particularly valuable in rare event prediction, as they consider intricate data patterns and enhance the overall quality of datasets. Researchers have applied a variety of imputation methods in the context of rare event prediction, yielding improved model performance and predictions [5,51,54,55,56,57,58,59]. For instance, Omar et al. [5] employed the MisForest algorithm [60], which is an iterative imputation method based on the Random Forest algorithm for the imputation of missing data. Cheon et al. [59] utilized advanced offset value approximation techniques involving predefined offset values for input data. Furthermore, Rafsunjani et al. [51] conducted an empirical investigation of various imputation techniques, including expectation maximization, mean imputation, soft impute, Multiple Imputation by Chained Equation (MICE), and Iterative SVD, in the context of APS failure prediction, highlighting MICE as a high-performance method in their predictions.

From the studied literature, it was seen that even though much research has employed data enrichment techniques in multiple domains, there is still a lack of applicability of these in the manufacturing domain. Given the scarcity, complexity, multidimensionality, and heterogeneity of manufacturing data [61], it is imperative to explore methods for improving predictive performance. Data enrichment techniques can play a significant role in this context. Exploring the synergistic potential of combining various data enrichment techniques presents a promising avenue for addressing the above rare event prediction challenges. This innovative approach represents a novel direction that has yet to be extensively explored within the field. By strategically utilizing these robust methodologies, we can harness the complete capabilities of the data and attain heightened levels of precision in manufacturing-based predictive analyses.

## 3. Methodology

The main objective of this research is to improve the detection and prediction of rare events in manufacturing by leveraging data enrichment approaches. Figure 1 shows a high-level depiction of the proposed experimental framework used in this investigation. The process involves primary data processing, followed by model development and selecting the best model for each dataset. Subsequently, secondary data processing (Data Processing II) is undertaken, incorporating the application of data enrichment techniques, culminating in comprehensive evaluations. A detailed flow diagram of the steps involved in the experimental setup is shown in Figure 2. Next, each of the steps involved in this setup will be elaborated in detail.

### 3.1. Datasets

This research employs multivariate numerical time series data, wherein records encompass the continuous recording of multiple interrelated data streams over a period of time. Considering Industry 4.0 as the domain of interest, we select five real-world datasets from different manufacturing sectors. These datasets can be categorized into two distinct types: naturally rare datasets and derived datasets. Naturally rare event datasets refer to datasets that inherently exhibit a low occurrence rate of specific events or phenomena [3]. Conversely, a derived dataset results from the transformation of an existing dataset that is initially not rare concerning the events, thereby creating rarity in the derived set [3]. As naturally rare datasets, we used the pulp-and-paper manufacturing dataset [10], the Bosch production line performance dataset [62], the Air Pressure System (APS) failure dataset [63,64], and the Future Factories (FF) dataset [65,66]. Derived datasets include the Ball-bearing dataset [67] and a sampled instance of the FF dataset. Among these datasets, only the pulp-and-paper manufacturing dataset and the FF dataset include a timestamp variable within their final datasets. Conversely, the remaining three datasets lack timestamp variables in their final configurations; however, they are organized chronologically. Consequently, the pulp-and-paper manufacturing dataset and the FF dataset were utilized specifically for rare event prediction, whereas all five datasets were experimented with for rare event detection. A detailed overview of these datasets is presented below.

#### 3.1.1. Pulp-and-Paper Manufacturing Dataset

The dataset has been collected from the pulp-and-paper industry [10] and includes data collected from several sensors placed in different parts of the machines in a paper mill along its length and breadth. These sensors measure metrics on raw materials (e.g., the amounts of pulp fiber, chemicals, etc.) and process variables (e.g., blade type, couch vacuum, rotor speed, etc.). The dataset has 18,398 records collected over 30 days. Data include sensor readings at regular time intervals (x’s), which are two minutes apart, and the event label (y). The records have the timestamp y, the binary response variable, and x1–x61, predictor variables. All the predictors are continuous variables except x28 and x61. x61 is a binary variable, and x28 is a categorical variable. There are only 124 rows with a value of y = 1, the rest have a value of y = 0. 1 denotes a sheet break.

#### 3.1.2. Bosch Production Line Performance Dataset

Bosch Inc. created the Bosch production line performance dataset [62] in 2016. This dataset represents a complex industrial system with four distinct segments and 52 individual workstations. Each of these workstations conducts various tests and measurements on specific components, yielding a dataset encompassing 4264 features. The dataset contains records of 1,183,747 individual parts, each associated with one of 4700 unique combinations. Most of the dataset consists of NULL values, with only 5% numeric values, 1% categorical values, and 7% of the timestamps being non-null. For the scope of our analysis, we only considered the dataset containing numerical values, which comprised 970 distinct features. To maintain consistency with the number of features in the pulp-and-paper dataset, we employed Random Forest feature importance scoring to identify and select the 59 most important variables. This dataset is highly imbalanced, and almost 99.5% of records are not facing any internal failure.

#### 3.1.3. Air Pressure System (APS) Failure Dataset

The dataset [63] contains data from Scania trucks, primarily focusing on the Air Pressure System (APS) responsible for tasks like braking and gear changes. It includes two classes: one for APS component failures (positive) and the other for failures not related to the APS (negative). The dataset comprises 60,000 examples in the training set (59,000 negative and 1000 positive, with a rarity percentage of 1.67%) and 16,000 examples in the test set, with 171 attributes per record. The numeric dataset represents a subset of all available data meticulously curated by domain experts. The attribute names are anonymized, consisting of numerical counters and histograms with open-ended conditions. The dataset has numerous missing values, with eight attributes having more than 50% missing values and only 2% of instances having complete data, while some cases exhibit up to 80% missing values. Consistent with the data preparation approach used in the Bosch dataset, we employed Random Forest feature importance scoring to identify and subsequently select the 59 most important variables.

#### 3.1.4. Ball-Bearing Dataset

The dataset [67] from Case Western Reserve University, Ohio, encompasses a range of components and parameters for analyzing ball bearings. The data were collected with a 2 hp motor equipped with accelerometers, capturing data on different fault sizes and types. For our research, we employed two data files for analysis. The first data file, *‘Normal_0.mat’*, comprises samples obtained under normal operating conditions from the normal baseline data. These samples were collected when the motor load was at zero horsepower (HP) and operating at approximately 1797 revolutions per minute (rpm). The faulty dataset utilized for our study includes samples from the outer race 12k Drive-End Bearing Fault Data, specifically the data file labeled *‘OR007@6_0.mat’*. These samples were selected due to an outer race fault and were obtained under similar conditions, with the motor operating at zero HP and an approximate speed of 1797 rpm. The original ball-bearing dataset does not exhibit rarity. Consequently, for our analysis, two distinct rare datasets were generated by maintaining the normal-to-rare ratios at 5% and 0.5%, and we used drive-end accelerometer data (DE) and fan-end accelerometer data (FE) as the features.

#### 3.1.5. Future Factories (FF) Dataset

The fourth dataset used in the study is the FF dataset, generated by the McNair Aerospace Center at the University of South Carolina in September 2023, specifically for applications in rocket assembly [65,66]. It originally encompassed data from runs lasting 3 h, 6 h, 7.5 h, and 24 h. Our investigation is focused on the six-hour data run. The comprehensive dataset includes various metrics such as conveyor VFD temperature, conveyor workstation statistics, cycle statistics (cycle count, cycle state, and cycle number), factory cell statistics (cabinet state and door statuses), material handling station statistics (relating to the picking of rocket parts), and data from four robots (load cell measurements, potentiometer data, and six different angles). The analysis within this study centers on two response variables, namely “Success” and “Anomaly,” with “Success” chosen as the primary response variable for our investigation. The initial frequency of the FF dataset was 10 Hz. A subsampled dataset was created by down-sampling the original data to a frequency of 1 Hz. After careful feature exploration and investigation, 12 features were taken for the modeling phase. As the second primary contribution of this paper, we are publishing the processed dataset we prepared from the six-hour run and the subsampled dataset.

The statistics of the datasets we used in the study are given in Table 1.

### 3.2. Methods for Rare Event Detection and Prediction

Each dataset was modeled using statistical, machine learning, and deep learning techniques for rare event detection and prediction, as detailed in Table 2. *Detection* involves the identification of events after they occur. *Prediction* is an ahead-of-time prediction of events, in which we used curve-shifting operations to shift the response labels in the data frame to facilitate the early prediction of an event. Within the scope of statistical approaches, the Autoregressive Integrated Moving Average (ARIMA) model was employed. Machine Learning methods involved the utilization of a Support Vector Machine (SVM), Adaboost, XGBoost, Weighted XGBoost, Random Forest, Logistic Regression, and Weighted Logistic Regression. For deep rearning, Simple Recurrent Neural Networks (RNNs), Convolutional Neural Networks (CNNs), Long Short-Term Memory (LSTM), an Autoencoder, and an LSTM Autoencoder were implemented. Following an evaluation of their outcomes, the best-performing model for each dataset was selected. Subsequently, employing the identical feature set for each of the datasets, the data enrichment process, which is described in Section 3.3, was conducted. The selected best-performing model was then applied to model these processed datasets.

Presently, there is a growing trend in the time series research community in developing transformer-based models for time series forecasting. However, there are certain limitations in using transformer-based models in time series like the loss of time series temporal information, the quadratic complexity of sequence length, the slow training and inference speed due to the encoder-decoder architecture, and overfitting issues [68,69,70]. Considering these limitations, we have not extended our research to explore transformer-based models as a baseline in our study.

### 3.3. Data Enrichment Approach

Three data enrichment techniques, data augmentation, sampling, and imputation, were executed on the pre-processed datasets as delineated in Table 3. While data augmentation and sampling techniques were applied across all five datasets, imputation was solely performed on the Bosch and APS datasets. The imputation technique was employed to address the substantial volume of missing values inherent in these particular datasets. A visual summary of the data enrichment approaches utilized has been depicted in Figure 3.

#### 3.3.1. Data Augmentation

We conducted several basic and advanced time-series-specific data augmentation methods, as listed in this section. For each dataset, best performing and weakly performing data augmentation methods were identified. In the context of time series forecasting, basic data augmentation methods encompass fundamental transformations employed on existing data to enhance the dataset [11]. These methods include techniques like cropping, adding noise, shifting, scaling, or rotating the time series to generate new instances. In contrast, advanced data augmentation methods involve more intricate techniques such as Seasonal Trend Decomposition, Generative Adversarial Networks (GANs), Variational Autoencoders (VAEs), Recurrent GANs (RGANs), etc. Ref. [11] applied these specifically to generate new synthetic instances while preserving the characteristics and patterns of the original time series data. This section delineates the techniques employed for data augmentation in this study, encompassing both basic and advanced techniques.

Basic augmentation techniques used:1.Relative change: Relative change calculation in time series data involves measuring the percentage change between consecutive data points, providing valuable information about the magnitude and direction of change over time. To calculate relative change, we compared the current value of a data point with its previous value, which is the value of the previous time step. The formula for relative change is as follows:
(1)$$\mathrm{Relative}\;\mathrm{change}=\left(\frac{\mathrm{Current}\;\mathrm{value}-\mathrm{Previous}\;\mathrm{value}}{\mathrm{Previous}\;\mathrm{value}}\right)*100$$The relative change in time series data, which can be positive or negative, indicates the rate and direction of change, helps identify patterns and trends, and highlights significant shifts or anomalies, thereby providing insights into important events or turning points.2.Lagged features: Lagged features incorporate past observations or time lags of the target variable or other relevant features into the dataset, enabling the model to consider historical behavior and dependencies over time. Lagged features were obtained by shifting the time series data by a certain number of time steps, and the shifted values were used as additional input features. This allowed the model to capture temporal dependencies and exploit autocorrelation.3.Rolling window: The rolling window technique generates overlapping windows of fixed size along the time series, computing summary statistics to capture short-term dynamics and variations. After empirical evaluations, we selected the optimal window size and calculated the mean values within each window. We used these as features for all input variables to enhance the model’s understanding of evolving trends.4.Expanding window: Expanding window augmentation gradually increases the size of the training data window over time, starting with a small initial window and progressively including more historical data. This technique enables the model to learn from longer-term dependencies, capturing overall trends and patterns that develop over a longer duration.5.Convolving: Convolving time series with a 1D kernel window aids in feature extraction and capturing local patterns, allowing the model to discern short-term dependencies and significant temporal features while reducing noise and highlighting essential information.6.Pooling: Pooling reduces the temporal resolution of time series without changing their length, effectively lowering dimensionality, computational time, and the risk of overfitting by minimizing parameters.7.Drifting: Drifting involves randomly and smoothly altering the values of the time series, with the extent determined by maximal drift and the number of drift points. It helps reveal long-term dependencies in the data.8.Time warping: Time warping randomly changes the speed of the timeline, controlled by the maximum speed ratio and the number of speed changes. This introduces variability and generates diverse time-based sequences. This enhances the model’s exposure to different temporal patterns, potentially improving its robustness and generalization capabilities.9.Quantizing: In quantizing, the time series values are controlled to a level set, or the values are rounded to the nearest level in the level set. Quantization can simplify complex data by reducing the number of distinct values. So, it can facilitate better pattern recognition and model training.10.Reversing: This reverses the timeline of the series. Reversing the series allows the model to learn patterns and relationships that might be prevalent in the original series but in reverse order. This technique is particularly significant in situations where there might be a symmetrical or cyclic nature within the data that could affect future predictions.11.Noising: Adding random noise to each time point of a time series independently increases variability, which improves the model’s generalizability, prevents overfitting, and enhances resilience to outliers.

Advanced augmentation techniques used:

As an advanced augmentation technique, seasonal trend decomposition analysis was conducted to decompose a time series into its fundamental components: its trend, seasonality, and residual. The main points of seasonal trend decomposition are as follows:1.Decomposition Involves separating a time series into its three aforementioned components. This decomposition helps in understanding the underlying patterns and structures in the data, enabling further analysis and modeling.2.Trend: The trend represents the long-term patterns of the time series data. It can capture the overall increase or decline in the series over an extended period.3.Seasonality: The seasonality includes repetitive patterns and fluctuations in a series that occur within a fixed time frame, like daily, weekly, monthly, or yearly cycles. Seasonality can be observed as regular and predictable patterns in the data.4.Residual: The residual represents the random and irregular fluctuations in the data that do not fall under the trend or seasonality components. It includes unexpected and unpredictable variations, measurement errors, or other factors. The residual can be obtained by subtracting the estimated trend and seasonality from the original data.

#### 3.3.2. Sampling

In our study, we experimented with the role of sampling in rare event prediction by employing three sampling techniques.

1.SMOTE-Tomek links: The SMOTE-Tomek Links method, proposed by Batista et al. in 2003 [71], combines the SMOTE (Synthetic Minority Oversampling Technique) and Tomek Links to improve class balance. The SMOTE generates synthetic samples for the minority class by creating new data points between existing ones. At the same time, Tomek Links remove majority class instances close to minority class instances, enhancing class separation. This approach involves generating synthetic minority samples and removing Tomek Links to achieve better class balance.2.SMOTE-ENN: The SMOTE-ENN method, developed by Batista et al. in 2004 [72], integrates the SMOTE and ENN (Edited Nearest Neighbors) techniques to enhance class balance. The SMOTE generates synthetic samples for the minority class, while ENN removes observations with differing class labels compared to their K-nearest neighbors. This method involves creating synthetic minority samples and subsequently applying ENN to eliminate mislabeled observations, ensuring balanced class proportions.3.ADASYN: The ADASYN algorithm, proposed by He et al. in 2008 [34], is designed to address the class imbalance by generating synthetic samples for the minority class based on their difficulty in classification. It focuses on harder-to-classify instances by assigning them higher weights, estimating local density distributions, and generating samples in areas of severe imbalance, thereby improving the representation of the minority class and enhancing classifier performance.

In general, the SMOTE-Tomek Links method proves advantageous when aiming to refine the majority class and diminish overlap with the minority class, particularly suited for datasets exhibiting distinct tomek links. Conversely, the SMOTE-ENN method is preferable when prioritizing noise reduction, actively eliminating noisy samples while simultaneously oversampling the minority class. ADASYN emerges as the optimal choice for datasets characterized by severe class imbalance and varying distributions within the minority class. Its adaptive approach adjusts the sampling strategy based on localized density information, ensuring a more nuanced representation of minority class instances. Most importantly, the selection of a sampling technique should align closely with specific dataset characteristics such as the severity of class imbalance, the presence of noise, and the nature of inter-class overlap, thereby enhancing the efficacy of rare event prediction models.

The term *“sampling techniques”* encompasses diverse methodologies across fields such as reliability analysis, including Monte Carlo simulation, importance sampling, and subset simulation. In contrast, our study focuses on SMOTE-Tomek Links, SMOTE-ENN, and ADASYN, specialized techniques in machine learning for addressing class imbalance and enhancing predictive model performance in scenarios involving rare events. In our experiments, the above sampling techniques were applied to the augmented and original datasets to compare the effects of augmentation on the final prediction.

#### 3.3.3. Imputation

Two methods were incorporated as imputation techniques. Replacing null values with zero was used as a simple imputation technique. An advanced imputation technique that involves rolling mean statistics for time series was proposed.

The steps involved in the advanced imputation method used are as follows.

1.Compute rolling mean statistics for each column within a specified window size (except for the ’time’ column).2.Calculate the means for the current window, the previous window (with a one-time step shift upwards), and the following window (with a one-time step shift downwards).3.Average the means obtained from the previous and subsequent windows and replace null or zero with the average mean.4.If either null or zero is identified, replace it based on the mean of the previous or subsequent window.5.For the last index in the data frame, if a column value is still null, update it with the mean of the previous window. (Note that if the first-row value is null, the initial computation and filling of the rolling mean and substituted values do not cover the first row’s missing or null value because the rolling window function requires a certain window size to generate meaningful statistical values. We took this assumption to preserve the temporal dependency of time series).

Section 3 described in detail the methodology followed in the research. The next section will be based on the evaluation of the data enrichment approaches and the results.

## 4. Evaluations, Results, and Discussion

This section evaluates the developed models in terms of performance measures and use cases, either detection or prediction. It also evaluates the influence of data enrichment techniques and compares experiments on real manufacturing data to extract dataset-specific novel insights. Finally, it includes an in-depth investigation and evaluation of the interpretability of model outcomes utilizing diverse methods and criteria. The experimental setup we designed is as follows.

1.Baseline Experiment [Section 4.2]: *Evaluate the baseline performance—Objective:* To identify the best-performing method for each dataset in rare event detection and prediction.2.Primary Experiment [Section 4.3]: *Evaluate the effect of data enrichment approaches*—Objective: To see how each data augmentation technique improves the overall performance in rare event detection and prediction.3.Secondary Experiment [Section 4.4]: *Evaluate the dataset-specific features towards model performance*—Objective: To identify the best configuration of input feature representation that yields better predictive performance in rare event detection and prediction.4.Tertiary Experiment [Section 4.5]: *Investigating the feature importance and model explainability aspect*—Objective: To explain the model predictions through feature importance in rare event prediction.

### 4.1. Performance Metrics

To assess the effectiveness of the best models, various performance metrics were employed, including precision, recall, F1-score, and accuracy. Due to the imbalanced nature of rare events, it is important to note that a higher accuracy score does not necessarily indicate a superior model. These are specifically suitable evaluation metrics in rare event prediction research due to their focus on the minority class, their ability to account for imbalanced datasets, the trade-off analysis between accuracy and sensitivity, their interpretability, and their robustness to class distribution changes [73]. Our objective is to develop a model that exhibits a tradeoff between high precision and recall metrics while also achieving a high level of prediction accuracy. The performance measures are defined in Equations (2)–(5).
(2)Recall=TP/(TP+FN)
(3)Precision=TP/(TP+FP)
(4)Accuracy=TP+TN/(TP+TN+FP+FN)
(5)F1score=2·Precision·Recall/(Precision+Recall)
where TP is the number of true positives, FP is the number of false positives, TN is the number of true negatives, and FN is the number of false negatives.

### 4.2. Baseline Experiment: Evaluate the Baseline Performance

Table 4 shows the best modeling technique for each dataset. As per the results, it can be seen that the weighted XGBoost model performed better in all the datasets for both rare event prediction and detection. The hyperparameters used in the best modeling techniques are included in Table A1 in Appendix A. Figure 4 shows the highest F1 scores of the baseline models for each model category.

In all the datasets, several parameters were optimized to maximize the performance of the Weighted XGBoost model and to prevent data overfitting. The optimal XGBoost hyperparameter values were selected after cross-validation, which included the number of iterations, max depth, subsample, lambda, alpha, and learning rate. Grid search cross-validation was used in fine-tuning the *scale_pos_weight* hyperparameter, which defines the ratio of the number of samples in negative classes to the positive class. It is a special hyperparameter designed in XGBoost to tune the behavior of the algorithm for imbalanced classification problems. Repeated stratified KFold cross-validation was used for the parameter tuning, where the performance metrics were optimized.

### 4.3. Primary Experiment: Evaluate the Effect of Data Enrichment Approaches

Table A2 in Appendix B and Table 5 show the evaluation results of the best modeling method with and without using data augmentation and sampling techniques in rare event detection and prediction, respectively. In Table 6, we show the best sampling method with and without data augmentation for rare event detection across all datasets. As per the results, it is seen that using augmentation as a data enrichment technique in detection has yielded better weighted average performance in normal and not-normal samples in the pulp-and-paper, APS, ball bearing, and FF datasets. For the Bosch dataset, using augmentation and Tomek link sampling has given better results. For rare event prediction, data augmentation resulted in high performance in both the pulp-and-paper and FF datasets. Table 7 and Figure 5 show the evaluation results of the best modeling method using simple and advanced imputation methods with data augmentation methods in rare event detection. From the results, it is clear that the advanced imputation method and data augmentation method has resulted in better performance metrics in rare event detection in APS and Bosch datasets.

The outcomes of the experiments reveal that the resampled FF dataset exhibits superior performance, even in the absence of data enrichment techniques. Several factors contribute to this improvement. First, the process of resampling can effectively reduce the noise within the dataset, resulting in a cleaner signal for the model to interpret. Second, the reduction in variability in the resampled data contributes to a more stable and consistent dataset, aiding in model generalization. Additionally, the resampling technique enhances the visibility of patterns, enabling the model to focus on more meaningful information and reducing the impact of short-term fluctuations. Finally, the resampling approach proves advantageous in capturing time series seasonality more effectively when operating at a lower frequency. These comprehensive benefits underscore the efficacy of the FF resampling method in enhancing the overall performance of the dataset for predictive modeling purposes.

### 4.4. Secondary Experiment: Evaluate the Dataset-Specific Features Towards Model Performance

We conducted several analysis experiments on the manufacturing datasets aiming to reveal dataset-specific novel insights. These experiments were unique for the dataset and were designed based on factors like dataset-specific features, data generation, rarity influence, and use cases, including detection or prediction, etc.

The unique evaluation criteria designed include the below set of experiments:1.Best shift period over data enrichment approach: We conducted this experiment on the rare event prediction use case based on the pulp-and-paper manufacturing and FF datasets. The aim was to find the best shift period or simply the best ahead-of-time prediction for these datasets. In the pulp-and-paper dataset, since there is a considerably high number of original features (59), experiments were conducted on four subsets of datasets: using original features (59) without any data enrichment approach, using 15 selected features using random forest feature importance, adding data augmentation to those 15 selected features, and using original features (59) with data augmentation techniques. With empirical evaluations with two-minute, four-minute, and six-minute shift periods, it was seen that using original features (59) with data augmentation and four-minute-ahead prediction yielded better macro average prediction results of both normal and rare classes in the pulp-and-paper manufacturing dataset, as seen in Table 8. In the FF dataset, experiments were conducted on two subsets of datasets; using original features (9) without any data enrichment approach and using original features (9) with data augmentation techniques. With empirical evaluations with one-minute, two-minute, and three-minute shift periods, it was observed that using original features (9) with data augmentation and one-minute ahead prediction yielded better prediction macro average results of both normal and rare classes in the FF dataset as seen in Table 9.2.Splitting method over data enrichment approach: This experiment was also conducted on the rare event prediction use case based on the pulp-and-paper manufacturing and FF datasets. The aim was to find the best splitting method that can be used in the best ahead-of-time prediction for these datasets. We analyzed the influence of the splitting method for final prediction over three splitting methods: random splitting, time-based splitting, and run-based splitting. Random splitting involved splitting original time series data into train and test splits randomly based on the ratio of 70:30. Time-based splitting involved taking the first 80% of the time as the training set and the remaining 20% as the test set. Run-based splitting involves dividing the split based on the actual running time of the machine. For the pulp-and-paper manufacturing dataset, all three splitting methods were used in experimentation. However, the run-based method was omitted in the FF dataset due to the uninterrupted operation of the data generation machine throughout the data collection process. For the pulp-and-paper manufacturing dataset, four sessions of running the machine were obtained by exploring the dataset, as included in Table 10. The overall results of this experiment are stated in Table 11 and Table 12, and it is seen that using data augmentation methods yielded better performance in all the splitting methods in both the datasets except for one session in the run-based splitting for the pulp-and-paper dataset.3.Influence of categorical features over the data enrichment approach: This experiment evaluated the influence of the categorical feature, *paper types* over data augmentation in rare event prediction in the pulp-and-paper dataset on time-based splitting. Time-based splitting was considered to be a good splitting method in this experiment since papers with the same paper types are usually manufactured in the same session. The pulp-and-paper dataset contains six different paper types, which are denoted in the *‘x28’* variable in the dataset. Experiments were conducted for the top three most-recorded paper types in the dataset. As per the results in Table 13, it can be seen that paper types and augmentation do not have a direct association with the performance of predicting a rare event.4.Rarity influence over data enrichment approach: The ball bearing dataset was used to create a derived dataset and to experiment with the influence of rarity over data augmentation. Two versions of the dataset, with 0.5% rarity and 5% rarity, were created for the derived datasets. Given the results in Table 14, we can say that the rarity does not directly affect the results of data augmentation in either case; data augmentation has led to better predictions.5.Best and weak augmentation methods across the datasets: Best and weak augmentation methods were selected for all five datasets using the *forward selection* feature selection algorithm. Here, we started with having no augmented feature in the model. Then, in each iteration, we kept adding the augmented features that best improved the performance of the best model. Those were considered the best augmentation methods. We identify merits and demerits associated with each data augmentation approach as summarized in Table A4 in Appendix D. If an addition of an augmented variable did not improve the performance of the model, they were considered weak augmentation methods. The results are shown in Table A3 in Appendix C. Here, the pulp-and-paper and FF datasets include the results of rare event prediction, whereas the Bosch, APS, and ball bearing datasets include the results of rare event detection.

### 4.5. Tertiary Experiment: Investigating the Feature Importance and Model Explainability Aspect

Given the results or outcomes of a detection or prediction, knowing and understanding the conclusion of a model is important. In our study, to understand the modeling behavior, we used two techniques to interpret the results given by the models. We experimented with these two techniques on the rare event prediction task. Firstly, we interpret the model results using the XGBoost feature importance technique, and secondly, we use Shapley Additive exPlanations (SHAP) values, proposed by Lundberg and Lee in 2017 [74], to interpret the outputs of the Weighted XGBoost model. This section includes the results of this investigation done on the pulp-and-paper manufacturing dataset.

1.Model Interpretations using XGBOOST feature importance:Here, firstly, a gradient-boosted tree model that includes a plot of the trees was created. Then, the *total cover* feature importance technique in the XGBoost algorithm was used to measure the importance or the total coverage of each feature during the construction of the XGBoost model. *The total cover* measure calculates the sum of the number of times a feature is used to split the data across all trees in the ensemble. In the XGBoost algorithm, the algorithm builds an ensemble of decision trees, where each tree tries to capture different patterns or relationships in the data. During the construction of these trees, the algorithm decides which features to use for splitting the data at each tree node. The ’total cover’ importance metric keeps track of the cumulative coverage of each feature across all the trees. A higher ’total cover’ value for a feature indicates that the feature has been selected for splitting frequently during the construction of the ensemble. This implies that the feature is considered important in distinguishing or explaining the target variable by the XGBoost model. After obtaining the gradient-boosted tree model and total cover measure, we compared the node names of the tree and total cover measure and realized they do match each other. An example for explanation is given in Figure 6, which includes both the above measures for predicting rare events in the pulp-and-paper manufacturing dataset with feature augmentation.2.Model Interpretations using SHAP—explainable AI with Shapley Additive exPlanations: SHAP (SHapley Additive exPlanations) [74] is an explainable AI technique rooted in Shapley values from cooperative game theory. Shapley values represent the mean marginal contribution of individual feature values within all possible values in the feature space. This method provides insight and an understanding of how each feature collectively contributes to the outcome predicted by black-box machine learning models. However, it has been found that SHAP values face limitations such as mathematical inconsistencies, failures to meet human-centric explainability goals, an inability to provide causal inference, variable factor contributions, and high computational complexity [75]. We employed three methods using SHAP: Global interpretation, Local interpretation, and Feature dependency analysis, to demonstrate feature analysis. In this section, we present the results of model interpretations based on the data augmentation enrichment technique for the pulp-and-paper manufacturing dataset.(i) Global interpretation. The SHAP values were calculated for every feature (augmented and real) and for the final dataset prepared to understand the feature’s importance and its impact on model output as in Figure 7. Then, the features were sorted in descending order based on their SHAP values and their importance in predicting rare events. The X-axis shows the impact of features on the model output. The color represents the average SHAP value of a feature at a position. Red regions have high-value features, and blue regions have low feature values. The vertical width of the color band represents the frequency of a particular SHAP value at a location. In this dataset, *cnv_x3* has the greatest impact on the model output. For the majorly visible variables, *cnv_x3* and *cnv_x2*, lower values of the feature, resulted in higher SHAP values, indicating a lower probability of a break occurring (negative correlation). When going down in the descending order of feature values, the SHAP values of these features are getting closer to zero, indicating a low impact on the model output. Some variables like *trend_x9*, *pool_x11*, and *lag_l_x3* have a mixed effect on model output. Figure 8 presents the average SHAP values for the features across all data samples in the augmented dataset. The average SHAP values for the features are sorted in descending order and show the global importance value of each feature on the model output. The average SHAP values demonstrate that *cnv_x3*, *cnv_x2*, *cnv_x42*, *tw_x44*, *expanding_mean_x47*, *trend_x29*, *drift_x23*, *lag_1_x19*, and *drift_x42* have the highest impact on the model output.(ii) Local interpretation. In Figure 9, we show the bar chart and the waterfall plot of the local features corresponding to the two selected samples. In Figure 9a(i,ii) we can see that *trend_x9*, *tw_x59*, *drift_x13*, *expanding_mean_x3*, and *lag_1_x3* (those in red) pushed the prediction towards being a break event. On the other hand, we see *cnv_x3*, *tw_x17*, *lag_1_x19*, and other features in the blue colour pushed the prediction towards the normal event. For observation b in Figure 9b(i,ii), *cnv_x3*, *cnv_x2*, *trend_x9*, and *tw_x59* (those in red) pushed the prediction towards being a break event. On the other hand, *expanding_mean_x47*, *tw_x17*, *pool_x11*, *lag_1_x3*, and other features which are in blue pushed the prediction towards the normal event. Comparison of a and b showed that for both observations, the features *trend_x9* and *tw_x59* increase the risk of a break event. However, we can see both of those variables having somewhat high percentages for the break class compared with the normal class. For example, the percentage of *tw_x59* was 4.114% for the break class; the impact was greater than the normal class, which has a percentage of 0.934%.(iii) Feature dependency analysis. Feature dependency analysis aims to explore the relationship between the features. A feature dependency plot includes the value of a particular feature on the x-axis and its SHAP value on the y-axis and is generated by changing a specific feature in the model. In the following, we present two examples where feature dependency analysis was analyzed.
*(a) Impact of cnv_x3 and expanding_mean_x17 on model output;*
In Figure 10a, *cnv_x3* was selected as the feature to determine its effect on rare event prediction when the *expanding_mean_x17* variable was changed. The red points represent the higher values of *expanding_mean_x17*, and the blue points represents lower values. It is seen that *cnv_x3* contains both positive and negative values, most negative values have positive SHAP values, and most positive values have negative SHAP values. So, lower values of the feature resulted in higher SHAP values; thus, there is a negative correlation between *cnv_x3* and the probability of a break occurring. For almost all the range of values in *cnv_x3* and its SHAP values, we can see *expanding_mean_x17* has high dependence.
*(b) Impact of tw_x44 and trend_x9 on model output;*
In Figure 10b, *tw_x44* was selected as the feature to determine its effect on rare event prediction when the *trend_x9* variable was changed. The red points represent the higher values of *trend_x9*, and the blue points represents lower values. It is seen that *tw_x44* contains both positive and negative values in the range of 3000 to −4000, and all these values have SHAP values that are negative or zero. It is seen that negative values of *tw_x44* having zero SHAP values and positive values of *tw_x44* have negative SHAP values. Higher values of the feature resulted in lower SHAP values; thus, there is a negative correlation between *tw_x44* and the probability of a break occurring. Clearly, there is a dependence between the two variables of *tw_x44* and *trend_x9*, as seen in b.

## 5. Conclusions

The detection and prediction of rare events in manufacturing hold the utmost importance in mitigating unintended occurrences and ensuring overall productivity in manufacturing use cases. This study deals with the detection and prediction of rare events in the manufacturing domain by implementing a framework that investigates the influence of data enrichment methods. Based on extensive empirical and ablation experiments on real-world datasets, we gain dataset-specific insights that are important for improving predictive accuracy. Furthermore, examining model interpretability using multiple techniques emphasizes the significance of comprehending the interpretive elements of predictive models in rare-event prediction. This study will lay the groundwork for future studies using more advanced and intelligent approaches to predict rare events that might happen in real-world manufacturing settings.

## Figures and Tables

**Figure 1 sensors-24-05009-f001:**
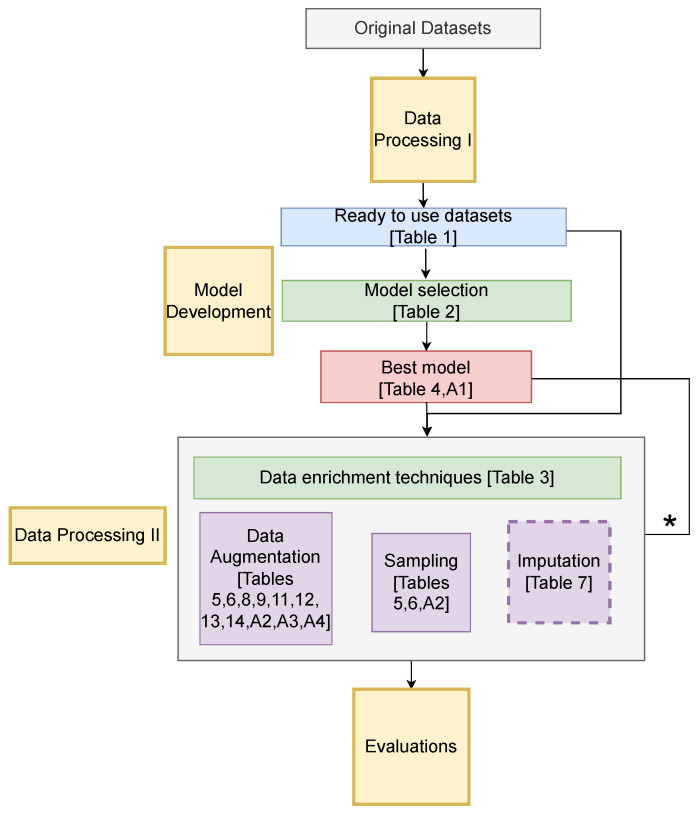
Experimental Setup. Data augmentation and sampling are applied across all datasets, as indicated by solid lines. Imputation is applied selectively to certain datasets, as denoted by dashed lines. * Indicates data enrichment techniques are applied iteratively multiple times.

**Figure 2 sensors-24-05009-f002:**
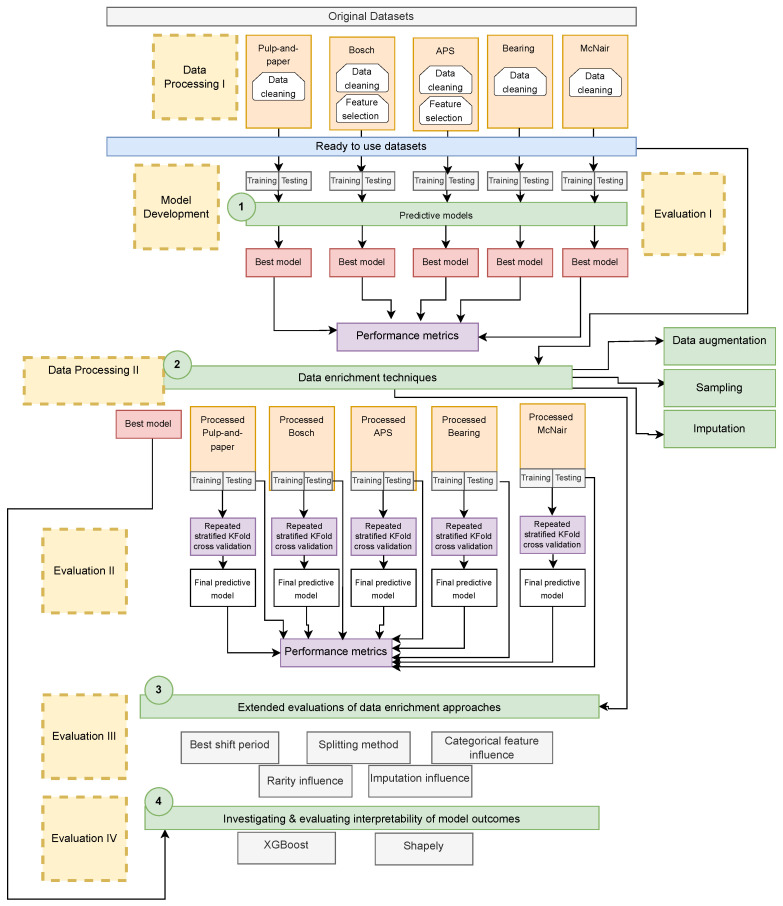
A detailed flow diagram of the steps involved in the proposed experimental setup.

**Figure 3 sensors-24-05009-f003:**
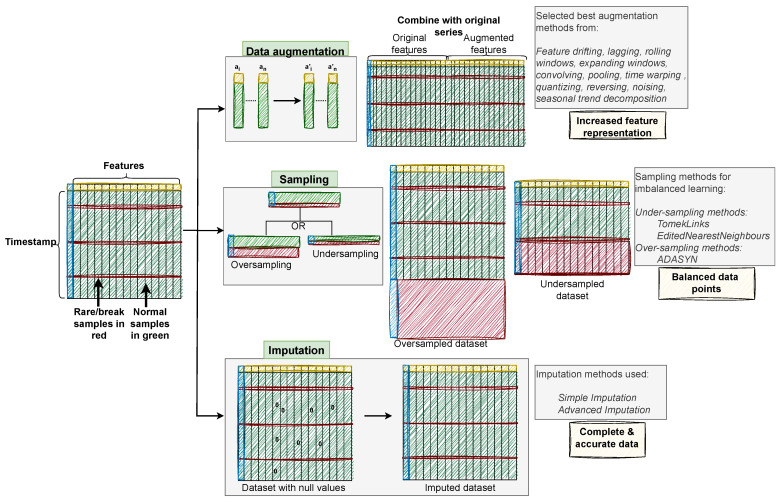
Visual Summary of the Data Enrichment Approach. As illustrated in the figure, data augmentation (shown in green) enhances the feature representation of the dataset. Sampling (shown in red) increases the number of data points (in oversampling) or reduces the number of data points (in undersampling) to balance the data distribution. Imputation addresses null values by replacing them with meaningful representative values, resulting in complete and accurate data (indicated by the absence of zeros).

**Figure 4 sensors-24-05009-f004:**
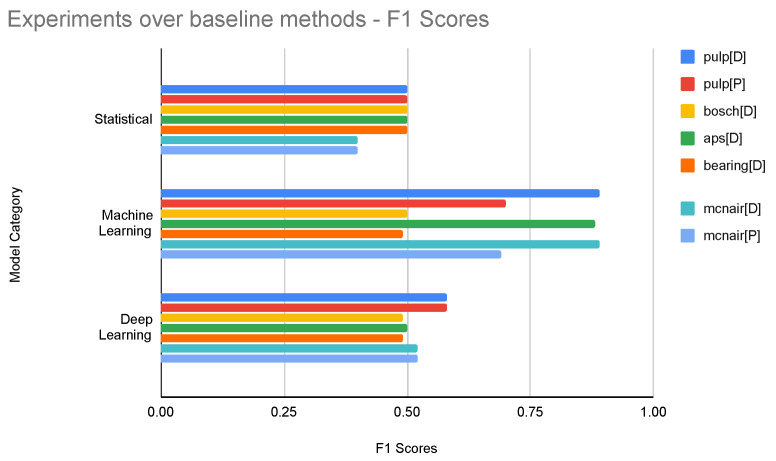
Experiments over baseline methods—F1 Scores.

**Figure 5 sensors-24-05009-f005:**
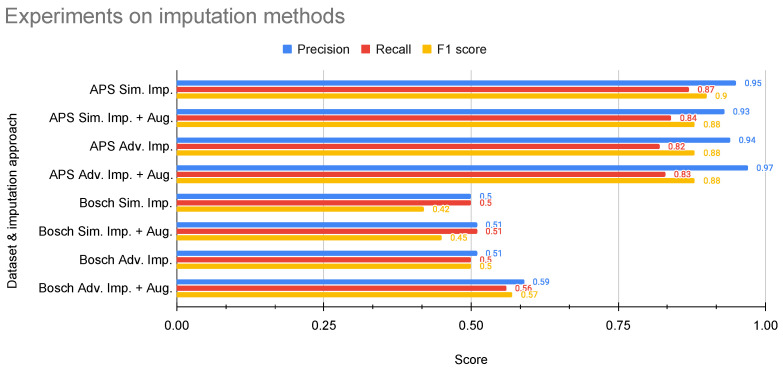
Experiments on simple and advanced imputation with augmentation methods in rare event detection.

**Figure 6 sensors-24-05009-f006:**
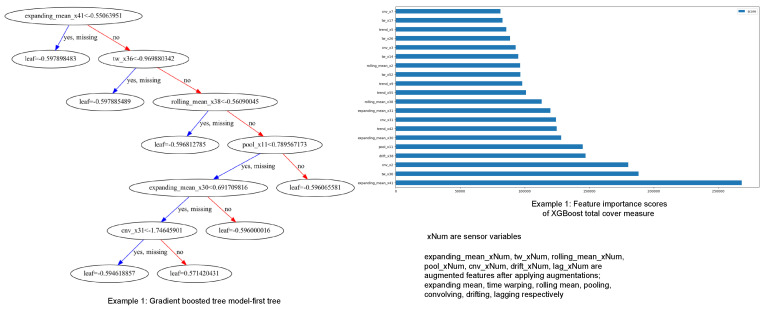
Example: Model Interpretations using XGBOOST feature importance.

**Figure 7 sensors-24-05009-f007:**
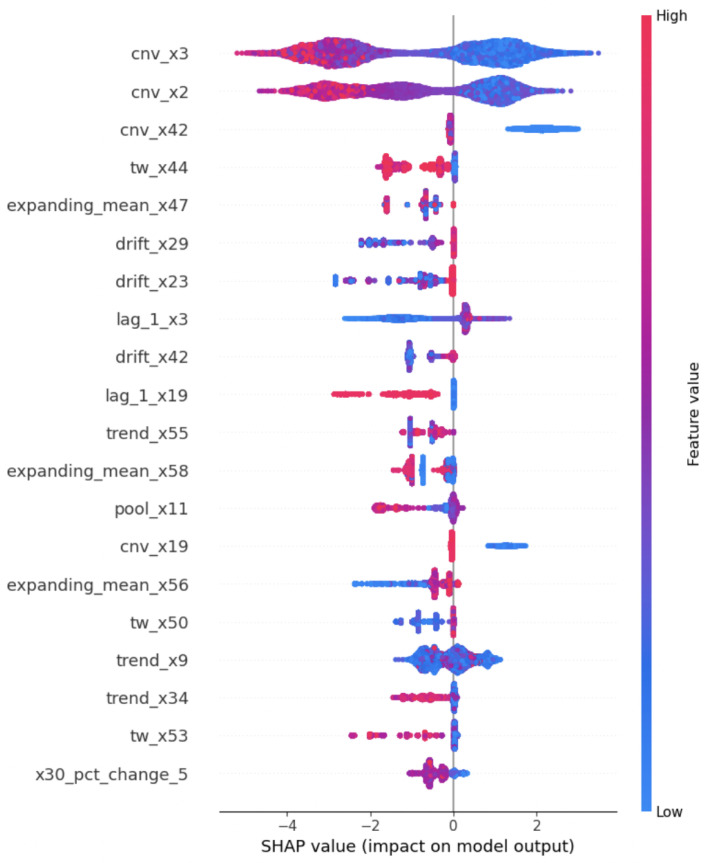
Summary of the SHAP values for the pulp-and-paper augmented dataset.

**Figure 8 sensors-24-05009-f008:**
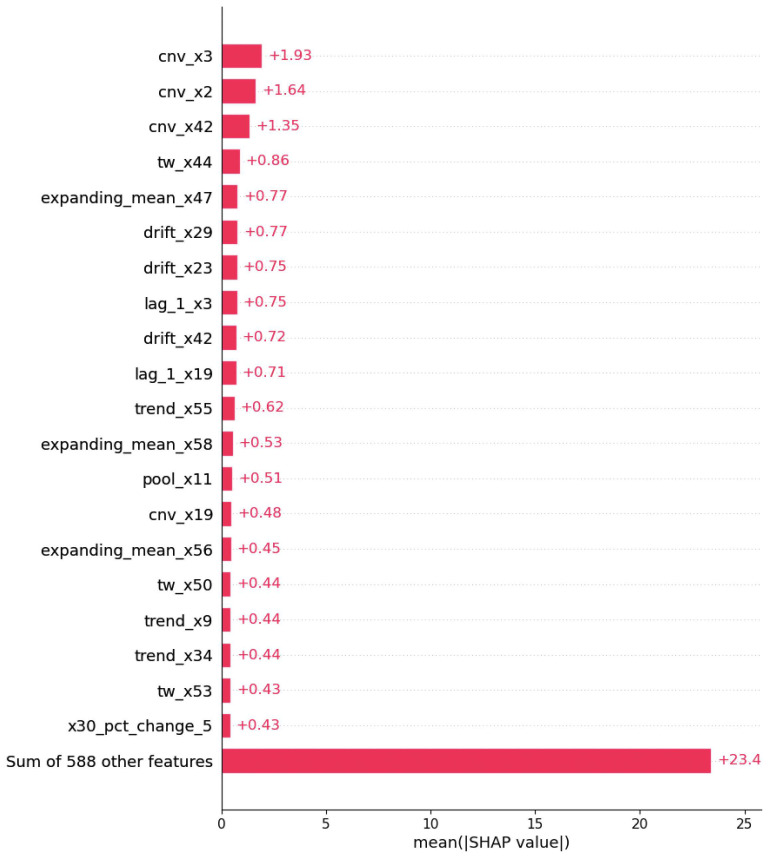
Average SHAP values of the features for the pulp-and-paper augmented dataset.

**Figure 9 sensors-24-05009-f009:**
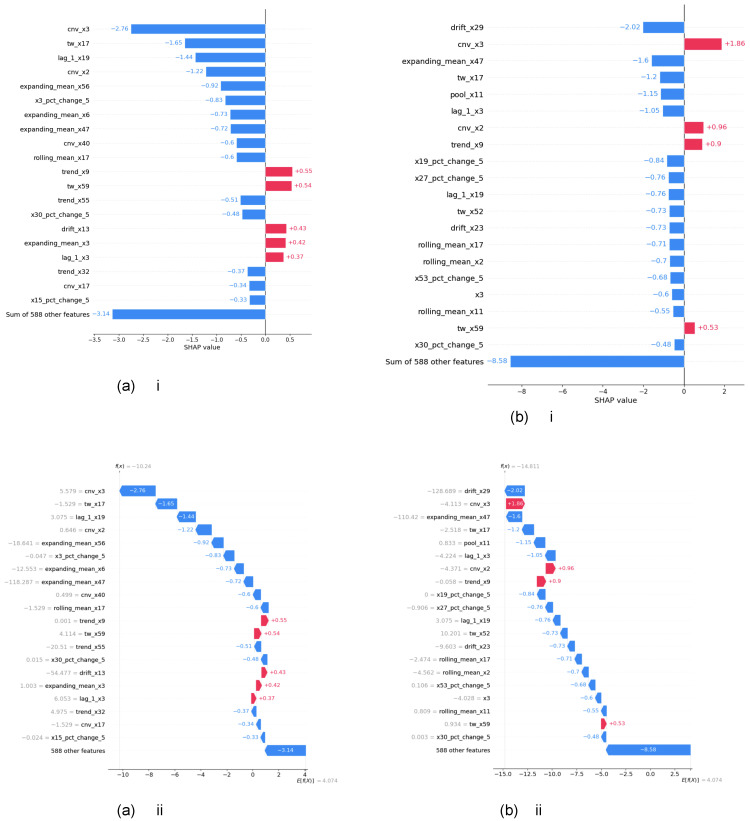
Bar chart (i) and Waterfall plot (ii) of the local effect of the features. (**a**) Observation 1342, a break event. (**b**) Observation 14, a normal event.

**Figure 10 sensors-24-05009-f010:**
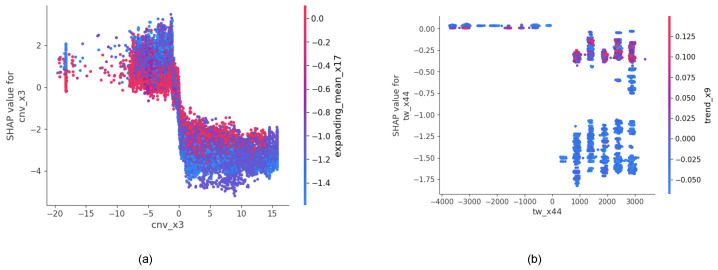
SHAP dependency analysis; (**a**) Impact of *cnv_x3* and *expanding_mean_x17* on model output; (**b**) Impact of *tw_x44* and *trend_x9* on model output.

**Table 1 sensors-24-05009-t001:** Datasets and their statistics.

Dataset Artifact	PP	BS	APS	FF-O	BR	FF-S
Rarity %	0.67%	0.58%	1.67%	1.70%	0.5%, 5%	1.70%
Original features	61	970	171	12	2	12
Selected features	59	59	59	12	2	12
No. of data points	18,398	1,183,747	60,000	216,034	20,000	21,604
Data collection period	30 days	Not mentioned	Not mentioned	6 h	Not mentioned	6 h
Industry type	Paper	Automotive	Automotive	Mechanical	Mechanical	Mechanical
Data sampled/not	Not	Not	Not	Not	Not	Yes
Sampling rate	-	-	-	-	-	1 Hz
Train/test split			70:30			

**Table 2 sensors-24-05009-t002:** Model categories and techniques.

Model Category	Techniques
Statistical	ARIMA
Machine Learning	SVM, Adaboost, XGBoost, Weighted XGBoost, Random Forest, Logistic Regression, Weighted Logistic Regression
Deep Learning	Simple RNN, CNN, LSTM, Autoencoder, LSTM Autoencoder

**Table 3 sensors-24-05009-t003:** Data enrichment approaches performed on different datasets.

Data Enrichment Technique	Pulp	Bosch	APS	Bearing	FF
Data augmentation	✓	✓	✓	✓	✓
Sampling	✓	✓	✓	✓	✓
Imputation	x	✓	✓	x	x

**Table 4 sensors-24-05009-t004:** Best performing models of all datasets.

	Statistical	ML	DL
**Precision**			
P and P [D]	ARIMA (0.5)	XGB (0.89)	CNN (0.60)
P and P [P]	ARIMA (0.5)	XGB (0.7)	CNN (0.62)
BS [D]	ARIMA (0.5)	XGB (0.51)	Autoencoder (0.53)
APS [D]	ARIMA (0.5)	XGB (0.94)	CNN (0.50)
BR (0.5%) [D]	ARIMA (0.5)	XGB (0.5)	CNN (0.50)
FF-O [D]	ARIMA (0.4)	XGB (0.89)	Autoencoder (0.52)
FF-O [P]	ARIMA (0.4)	XGB (0.8)	Autoencoder (0.52)
**Recall**			
P and P [D]	ARIMA (0.5)	XGB (0.89)	CNN (0.56)
P and P [P]	ARIMA (0.5)	XGB (0.7)	CNN (0.56)
BS [D]	ARIMA (0.5)	XGB (0.50)	Autoencoder (0.54)
APS [D]	ARIMA (0.5)	XGB (0.82)	CNN (0.50)
BR (0.5%) [D]	ARIMA (0.5)	XGB (0.49)	CNN (0.50)
FF-O [D]	ARIMA (0.4)	XGB (0.89)	Autoencoder (0.54)
FF-O [P]	ARIMA (0.4)	XGB (0.65)	Autoencoder (0.54)
**F1 Score**			
P and P [D]	ARIMA (0.5)	XGB (0.89)	CNN (0.58)
P and P [P]	ARIMA (0.5)	XGB (0.7)	CNN (0.58)
BS [D]	ARIMA (0.5)	XGB (0.50)	Autoencoder (0.53)
APS [D]	ARIMA (0.5)	XGB (0.88)	CNN (0.50)
BR (0.5%) [D]	ARIMA (0.5)	XGB (0.49)	CNN (0.50)
FF-O [P]	ARIMA (0.4)	XGB (0.89)	Autoencoder (0.52)
FF-O [P]	ARIMA (0.4)	XGB (0.69)	Autoencoder (0.52)
**Support**			
P and P [D,P]	3655 (3608 + 47)		
BS [D]	236,750 (235,396 + 1354)		
APS [D]	12,000 (11,816 + 184)		
BR (0.5%) [D]	4000 (3983 + 17)		
FF-O [D,P]	42,460 (42,407 + 53)		
**Accuracy**			
P and P [D]	ARIMA (0.99)	XGB (1)	CNN (0.98)
P and P [P]	ARIMA (0.99)	XGB (0.98)	CNN (0.98)
BS [D]	ARIMA (0.5)	XGB (0.99)	Autoencoder (0.97)
APS [D]	ARIMA (0.5)	XGB (0.99)	CNN (0.50)
BR (0.5%) [D]	ARIMA (0.5)	XGB (0.97)	CNN (0.9)
FF-O [D]	ARIMA (0.4)	XGB (0.99)	Autoencoder (0.7)
FF-O [P]	ARIMA (0.4)	XGB (0.95)	Autoencoder (0.79)

P and P—pulp-and-paper, BS—Bosch, APS—Air Pressure Systems, BR—Ball bearing, FF—S-Future factories sampled, FF—O-Future factories original, D—Detection, P—Prediction, ML—Machine learning, DL—Deep learning.

**Table 5 sensors-24-05009-t005:** Data augmentation and sampling results across all datasets: rare event prediction.

PM	Dataset	All (Aug. + Tomek Samp.)	All (Aug. + ENN Samp.)	All (Aug. + ADASYN Samp.)	Aug. Only	Samp. Only-tomek	Samp. Only-ENN	Samp. Only-ADASYN	No Method
P	P and P	0.87	0.72	0.88	**0.9**	0.86	0.6	0.7	0.7
	FF-O								**1**
R	P and P	0.72	0.74	0.7	**0.78**	0.73	0.64	0.63	0.7
	FF-O	**1**	1	**1**	**0.98**	0.85	0.91	0.87	0.67
F1	P and P	0.78	0.73	0.76	**0.83**	0.77	0.62	0.65	0.7
	FF-O	**1**	0.97	**0.99**	**0.99**	0.76	0.73	0.76	0.73
S	P and P	3655 (3608 + 47)							
	FF-O	42,460 (42,407 + 53)							
A	P and P	0.99	0.99	0.99	**0.99**	0.99	0.98	0.99	0.98
	FF-O	**1**	1	**1**	**1**	1	1	1	1

P and P—pulp-and-paper, BS—Bosch, APS—Air Pressure Systems, BR—Ball bearing, FF—S-Future factories sampled, FF—O-Future factories original, PM—Performance metric, P—Precision, R—Recall, F1—F1 Score, A—Accuracy, S—Support, Aug.—Augmentation, Samp.—Sampling. Bold numbers indicate the best performance.

**Table 6 sensors-24-05009-t006:** Best sampling methods with and without data augmentation across all datasets: rare event detection.

		All (Aug. + Tomek Samp.)			Aug. Only			Samp. Only-Tomek			No Method		
	**Support**	**P**	**R**	**F1**	**P**	**R**	**F1**	**P**	**R**	**F1**	**P**	**R**	**F1**
P and P	3655	0.82	0.94	0.87	**0.92**	**0.92**	**0.92**	0.81	0.92	0.86	0.89	0.89	0.89
BS	236,750	**0.82**	**0.52**	**0.53**	0.59	0.56	**0.57**	0.5	0.5	0.5	0.51	0.5	0.5
APS	12,000	0.84	0.82	0.83	**0.97**	0.83	**0.88**	0.85	0.9	0.87	0.94	0.82	0.88
BR	4000	0.89	1	0.94	**0.97**	**1**	**0.97**	0.5	0.54	0.39	0.5	0.49	0.49
FF-S	45										0.97	0.92	0.94
FF-O	42,460	**1**	**1**	**1**	**0.96**	**0.99**	**0.98**	0.86	0.92	0.89	0.89	0.89	0.89

P and P—pulp-and-paper, BS—Bosch, APS—Air Pressure Systems, BR—Ball bearing, FF—S-Future factories sampled, FF—O-Future factories original, P—Precision, R—Recall, F1—F1 Score, A—Accuracy, Aug.—Augmentation, Samp.—Sampling. Bold numbers indicate the best performance.

**Table 7 sensors-24-05009-t007:** Imputation influence over data enrichment approach.

Dataset	Imputation Method	Precision	Recall	F1 Score	Accuracy	Support
APS	Sim. Imp.	0.95	0.87	0.9	0.99	12,000 (11,816 + 184)
	Sim. Imp. + Aug.	0.93	0.84	0.88	0.99	
	Adv. Imp.	0.94	0.82	0.88	0.99	
	**Adv. Imp. + Aug.**	**0.97**	**0.83**	**0.88**	**0.99**	
Bosch	Sim. Imp.	0.5	0.5	0.42	0.7	236,750 (235,396 +1354)
	Sim. Imp. + Aug.	0.51	0.51	0.45	0.8	
	**Adv. Imp.**	**0.51**	**0.5**	**0.5**	**0.99**	
	**Adv. Imp. + Aug.**	**0.59**	**0.56**	**0.57**	**0.99**	

Sim. Imp.—Simple imputation, Adv. Imp.—Advanced imputation, Aug.— Augmentation. Bold numbers indicate the best performance.

**Table 8 sensors-24-05009-t008:** Shift period vs. data augmentation—pulp-and-paper dataset.

	Precision			Recall			F1 Score			Accuracy		
**Support**	**3655 (3608 +47)**											
Number of mins. ahead	2	4	6	2	4	6	2	4	6	2	4	6
No Aug., all features	0.5	0.7	0.83	0.5	0.7	0.6	0.5	0.7	0.65	0.99	0.98	0.98
Selected features	0.53	0.65	0.83	0.52	0.68	0.59	0.52	0.66	0.63	0.99	0.99	0.98
Selected features + Aug.	0.62	0.77	0.74	0.52	0.68	0.54	0.53	0.71	0.57	0.99	0.99	0.98
**Aug.**	0.5	**0.9**	**0.9**	0.5	**0.78**	0.67	0.5	**0.83**	**0.73**	0.99	**0.99**	0.99

Aug.—Augmentation. Bold numbers indicate the best performance.

**Table 9 sensors-24-05009-t009:** Shift period vs. data augmentation—FF dataset.

	Precision			Recall			F1 Score			Accuracy		
**Support**	**42,460 (42,407 + 53)**											
Number of mins. ahead	1	2	3	1	2	3	1	2	3	1	2	3
No Aug., All features	**0.86**	0.89	0.81	**0.67**	0.64	0.63	**0.73**	0.71	0.68	**1**	1	1
All features + Aug.	1	**1**	**1**	0.98	**0.99**	**0.99**	0.99	**1**	**1**	1	**1**	**1**

Aug.—Augmentation. Bold numbers indicate the best performance.

**Table 10 sensors-24-05009-t010:** Run-based splitting statistics—pulp-and-paper dataset.

Session	Start Time to End Time	Rare Event Count	Normal Event Count	Delay to Start Next Session
Session 1	5/1/99 0:00 to 5/7/99 12:36	37	4140	52 mins.
Session 2	5/7/99 13:28 to 5/18/99 4:22	38	7288	13 h 54 mins.
Session 3	5/18/99 18:16 to 5/24/99 23:04	27	4114	1 h 12 mins.
Session 4	5/25/99 0:16 to 5/29/99 0:06	22	2732	Process ended
Total samples		*124*	*18,274*	

**Table 11 sensors-24-05009-t011:** Splitting method vs. data augmentation—pulp-and-paper dataset.

Splitting Method	w/o Aug. or w Aug.	Precision	Recall	F1 Score	Accuracy	Support
Random	w/o Aug.	0.7	0.7	0.7	0.99	3655 (3608 + 47)
	**w Aug.**	**0.9**	**0.78**	**0.83**	**0.99**	3655 (3608 + 47)
Time based	w/o Aug.	0.52	0.5	0.51	0.98	3655 (3608 + 47)
	**w Aug.**	**0.79**	**0.54**	**0.57**	**0.98**	3655 (3608 + 47)
Run based	**session 1—w/o Aug.**	**1**	**0.79**	**0.87**	**0.99**	828 (816 + 12)
	session 1—w Aug.	0.85	0.79	0.82	0.99	828 (816 + 12)
	session 2—w/o Aug.	0.81	0.66	0.71	0.99	1458 (1442 + 16)
	**session 2—w Aug.**	**1**	**0.75**	**0.83**	**0.99**	1458 (1442 + 16)
	session 3—w/o Aug.	0.75	0.75	0.75	0.99	823 (815 + 8)
	**session 3—w Aug.**	**1**	**0.88**	**0.93**	**1**	823 (815 + 8)
	session 4—w/o Aug.	0.99	0.57	0.62	0.99	547 (540 + 7)
	**session 4—w Aug.**	**0.78**	**0.78**	**0.78**	**0.99**	547 (540 + 7)

w/o Aug.—without augmentation, w Aug.—with augmentation. Bold numbers indicate the best performance.

**Table 12 sensors-24-05009-t012:** Splitting method vs data augmentation—FF dataset.

Splitting Method	w/o Aug. or w Aug.	Precision	Recall	F1 Score	Accuracy	Support
Random	w/o Aug.	0.86	0.67	0.73	1	42,460 (42,407 + 53)
	**w Aug.**	**1**	**0.98**	**0.99**	**1**	
Time based	w/o Aug.	0.72	0.55	0.58	1	
	**w Aug.**	**0.72**	**0.7**	**0.71**	**1**	

w/o Aug.—without augmentation, w Aug.—with augmentation. Bold numbers indicate the best performance.

**Table 13 sensors-24-05009-t013:** Influence of categorical features(ex: paper types) over data enrichment approach—pulp-and-paper dataset for time-based splitting.

Paper Type	w/o Aug. or w Aug	Precision	Recall	F1 Score	Accuracy	Support
96	w/o Aug.	0.49	0.5	0.5	0.98	1301 (1285 + 16)
	**w Aug.**	**0.99**	**0.53**	**0.56**	**0.99**	
82	w/o Aug.	0.5	0.5	0.5	0.99	876 (872 + 4)
	w Aug.	0.5	0.5	0.5	0.99	
118	w/o Aug.	0.5	0.5	0.5	0.99	527 (523 + 4)
	w Aug.	0.5	0.49	0.49	0.97	

w/o Aug.—without augmentation, w Aug.—with augmentation. Bold numbers indicate the best performance.

**Table 14 sensors-24-05009-t014:** Rarity influence over data enrichment approach—Ball-bearing dataset.

Rarity %	w/o Aug. or w Aug.	Precision	Recall	F1 Score	Accuracy	Support
5%	w/o Aug.	0.5	0.48	0.32	0.4	4000 (3802 + 17)
	**w Aug.**	**0.96**	**1**	**0.98**	**1**	
0.50%	w/o Aug.	0.5	0.49	0.49	1	4000 (3983 + 17)
	**w Aug.**	**0.97**	**1**	**0.99**	**1**	

w/o Aug.—without augmentation, w Aug.—with augmentation. Bold numbers indicate the best performance.

## Data Availability

The code and data required to reproduce the results are available at https://github.com/ChathurangiShyalika/Rare_Event_Analysis (accessed on 25 June 2024). We also utilized publicly available datasets in our work, which are included and cited in the Datasets Section 3.1. We confirm that no copyright issues are associated with these datasets, and we ensured the proper referencing and citation of all sources.

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
