# Peer review of "Evaluating the Role of Data Enrichment Approaches towards Rare Event Analysis in Manufacturing"

_sensors, 2024, doi:10.3390/s24155009_

Round 1

Reviewer 1 Report

Comments and Suggestions for Authors

This paper investigates the role of three data enrichment techniques, i.e., data augmentation, sampling, and imputation, for rare event detection and prediction in manufacturing. The paper is overall well-written and well-organized. To improve the manuscript, the reviewer would like to request the authors to address the following issues:

1. The authors enumerate multiple basic augmentation techniques, but the differences among these techniques are not clear. It is desirable a sum-up table should be given to better compare the pros and cons of these techniques. Besides, the authors should explicitly state which techniques are used in this study.

2. Similarly, the authors should use a summarized paragraph to discuss the differences among the three sampling techniques, and point out which technique should be used in which case. Furthermore, the phrase “sampling techniques” is also widely used in other community, e.g., the reliability community. In those communities, popular sampling techniques are Monte Carlo simulation, importance sampling, and subset simulation. The authors should also briefly clarify the difference between these sampling techniques and the sampling techniques mentioned in the paper.

Comments on the Quality of English Language

The English quality is fine.

Author Response

Thank you very much for the review, comments, and suggestions. We have addressed each point in the revised manuscript and provided explanations below.

Review 1: The authors enumerate multiple basic augmentation techniques, but the differences among these techniques are not clear. It is desirable a sum-up table should be given to better compare the pros and cons of these techniques. Besides, the authors should explicitly state which techniques are used in this study.

Comment 1:  

  • The Basic Augmentation Techniques section in section 3.3.1 includes the techniques used in our study. We have included descriptions of each technique and their differences in the section. It should be noted that all these basic techniques were incorporated into all the datasets, and Table A.3 depicts the performance of each dataset across all these basic techniques.
  • We included a table in Appendix D based on the analysis of each data augmentation technique and linked to the main text in lines 686-687.

Review 2: Similarly, the authors should use a summarized paragraph to discuss the differences among the three sampling techniques, and point out which technique should be used in which case.

Comment 2: 

Added a summarized paragraph to compare each of the three techniques in lines 506-516.

Review 3: Furthermore, the phrase “sampling techniques” is also widely used in other community, e.g., the reliability community. In those communities, popular sampling techniques are Monte Carlo simulation, importance sampling, and subset simulation. The authors should also briefly clarify the difference between these sampling techniques and the sampling techniques mentioned in the paper.

Comment 3:

  • Incluced this in lines 517-521.

Reviewer 2 Report

Comments and Suggestions for Authors

The authors investigate the role of data enrichment techniques combined with supervised machine-learning techniques for rare event detection and prediction. They use time series data augmentation and sampling strategies to address data scarcity, maintaining its patterns, and imputation techniques, and evaluate and investigate 15 machine learning models. The paper is well-written, and the results of numerical investigations make the investigation of the research seem promising. The reviewer believes that the paper needs improvement in the following issues:

- In Section 3.1, the description of the rarity criteria for some of the datasets appears to be missing.

- In the rare event detection method in Secton 3.2, the model selection can be very sensitive to the choice of hyperparameters of the machine- or deep- learning algorithms (e.g., the choice of parameters in SVM, the choice of number of layers in LSTM, etc.). The reviewer believes that the authors need to provide hyperparameter values or provide additional investigates about this point.

- Similarity, lines 560-561, the authors present the results for the best modeling technique for each dataset. Could the authors elaborate on the tunning process for each algorithm?

- It would be helpful to the reader to express a simple definition of the criteria (precision, recall, F1, etc.) used on line 551.

- Could the authors elaborate on the meaning of the numbers in Table 4 (e.g., parenthesis, addition)? And Also Table 5, 7, etc?

Author Response

Thank you very much for the review, comments, and suggestions. We have addressed each point in the revised manuscript and provided explanations below.

Review 1- In Section 3.1, the description of the rarity criteria for some of the datasets appears to be missing.

Comment 1:

  • Fixed missing descriptions of rarity for two datasets in lines 266-267  and 272-273.

Review 2:

In the rare event detection method in Secton 3.2, the model selection can be very sensitive to the choice of hyperparameters of the machine- or deep- learning algorithms (e.g., the choice of parameters in SVM, the choice of number of layers in LSTM, etc.). The reviewer believes that the authors need to provide hyperparameter values or provide additional investigates about this point.

Comment 2:

  • Added a table to present the tuned hyperparameters used in the best models in Appendix Section A.1

Review 3: Similarity, lines 560-561, the authors present the results for the best modeling technique for each dataset. Could the authors elaborate on the tunning process for each algorithm?

Comment 3:

  • Added a table to present the tuned hyperparameters used in the best models in Appendix Section A.1

Review 4: It would be helpful to the reader to express a simple definition of the criteria (precision, recall, F1, etc.) used on line 551.

Comment 4:

  • Added in lines 575-578

Review 5:

Could the authors elaborate on the meaning of the numbers in Table 4 (e.g., parenthesis, addition)? And Also Table 5, 7, etc?

Comment 5:

  • This has already been added to the table descriptions.